# A Comprehensive Review of Recent Research into the Effects of Antimicrobial Peptides on Biofilms—January 2020 to September 2023

**DOI:** 10.3390/antibiotics13040343

**Published:** 2024-04-09

**Authors:** Alessio Fontanot, Isabella Ellinger, Wendy W. J. Unger, John P. Hays

**Affiliations:** 1Department of Medical Microbiology & Infectious Diseases, Erasmus University Medical Centre (Erasmus MC), Dr. Molewaterplein 40, 3015 GD Rotterdam, The Netherlands; a.fontanot@erasmusmc.nl (A.F.); w.unger@erasmusmc.nl (W.W.J.U.); 2Department of Pediatrics, Laboratory of Pediatrics, Erasmus University Medical Center Rotterdam, Sophia Children’s Hospital, Dr. Molewaterplein 40, 3015 GD Rotterdam, The Netherlands; 3Institute of Pathophysiology and Allergy Research, Center for Pathophysiology, Infectiology and Immunology, Medical University of Vienna, Währinger Gürtel 18–20, 1090 Vienna, Austria; isabella.ellinger@meduniwien.ac.at

**Keywords:** biofilm, antimicrobial peptides, AMPs, *Pseudomonas aeruginosa*, *Staphylococcus aureus*, LL-37, polymyxin B, GH12, Nisin

## Abstract

Microbial biofilm formation creates a persistent and resistant environment in which microorganisms can survive, contributing to antibiotic resistance and chronic inflammatory diseases. Increasingly, biofilms are caused by multi-drug resistant microorganisms, which, coupled with a diminishing supply of effective antibiotics, is driving the search for new antibiotic therapies. In this respect, antimicrobial peptides (AMPs) are short, hydrophobic, and amphipathic peptides that show activity against multidrug-resistant bacteria and biofilm formation. They also possess broad-spectrum activity and diverse mechanisms of action. In this comprehensive review, 150 publications (from January 2020 to September 2023) were collected and categorized using the search terms ‘polypeptide antibiotic agent’, ‘antimicrobial peptide’, and ‘biofilm’. During this period, a wide range of natural and synthetic AMPs were studied, of which LL-37, polymyxin B, GH12, and Nisin were the most frequently cited. Furthermore, although many microbes were studied, *Staphylococcus aureus* and *Pseudomonas aeruginosa* were the most popular. Publications also considered AMP combinations and the potential role of AMP delivery systems in increasing the efficacy of AMPs, including nanoparticle delivery. Relatively few publications focused on AMP resistance. This comprehensive review informs and guides researchers about the latest developments in AMP research, presenting promising evidence of the role of AMPs as effective antimicrobial agents.

## 1. Introduction

A biofilm is a complex structure formed by surface-attached, or non-surface attached, aggregates of microorganisms that in nature provide protection from desiccation, shear stress, toxic compounds, and protozoan grazing [1]. In the clinical environment, biofilms are associated with chronic and inflammatory diseases, as well as resistance to antibiotic therapy—microbial populations within biofilms tend to include minimally metabolically active ‘persister’ and/or ‘viable, but nonculturable’ (VBNC) sub-populations of bacteria [2]. Biofilms comprise an extracellular matrix (ECM), containing a range of extracellular polymeric substances (EPS), including cellular debris, proteins, polysaccharides, lipids, and nucleic acids [3], and the formation of biofilms relies on key signaling molecules such as c-di-GMP, proteins, and small regulatory RNAs (sRNAs), as well as quorum sensing (QS) [4]. Microscopic and biochemical techniques help in morphological analysis and comprehending the formation and variability of biofilms [5].

The emergence of global antimicrobial resistance (AMR) as a significant global health crisis is exacerbated by biofilm formation in One Health environments and a diminishing supply of novel antibiotics. Antimicrobial peptides (AMPs) are promising for fighting multidrug-resistant bacteria and biofilm formation [6,7]. AMPs have broad-spectrum activity against various pathogens through several inhibition mechanisms [8]. They can be categorized by structure, function, and target, and their reactivity relies on determinants such as cationic charge, beta fold, amphipathicity, and hydrophobicity [9,10]. AMPs and their potential role in preventing biofilm-related infections could form an important strategy in the fight against global pandemic AMR. Research in this broad scientific area is being constantly updated. Therefore, this comprehensive review collects and categorizes the latest research (from January 2021 to September 2023) in this fast-moving field and provides an easy-to-read overview of current progress, including a brief overview of the future direction of this research.

In this respect, the following databases were initially searched from inception until 28 September 2023: Embase; Medline ALL; Web of Science Core Collection, i.e., the Science Citation Index Expanded (1975–present), the Social Sciences Citation Index (1975–present), the Arts & Humanities Citation Index (1975–present), the Conference Proceedings Citation Index-—Science (1990–present), the Conference Proceedings Citation Index—Social Science & Humanities (1990–present), and the Emerging Sources Citation Index (2005–present); the Cochrane Central Register of Controlled Trials; and Google Scholar (searched via ‘Publish or Perish’). Database search terms included ‘polypeptide antibiotic agent’ (the preferred term for antimicrobial peptides in Embase), ‘antimicrobial peptide’ and ‘biofilm’ according to the following search criteria: (‘polypeptide antibiotic agent’/mj/de OR (((antimicrobial* OR anti-microbial*) NEAR/3 (peptide*))):ti) AND (biofilm/mj/exp OR (biofilm* OR bio-film*):ti) NOT (review/exp OR (review):ti) NOT ([Conference Abstract]/lim OR [Conference Review]/lim) AND [ENGLISH]/lim.

In order to limit the number of articles, searches were restricted to the publication title and major index terms. Reviews and conference abstracts were excluded and only publications in English were used. The results were then collated, and duplicates were removed.

The initial search (from database inception to 28 September 2023) was intended to provide the authors with an overview of the number of references available for this comprehensive review. Based on this information, a decision was then made to focus on publications meeting the search criteria that spanned the years January 2020–September 2023, which included a final total of 150 publications. This total of 150 publications was considered sufficient to provide an informative comprehensive review of the field of recent (in the previous three years) AMP-associated biofilm research.

After collating these 150 published titles, the publications were manually sorted into seven distinct categories based on a ‘weighting’ of publication titles (Figure 1). The highest weighting was reserved for research into the use of AMP combinations as this research requires collaboration between multiple scientific fields, including, for example, chemistry, materials science, clinical medicine, and microbiology. Investigations into AMP delivery efficiency are crucial for potentially overcoming the challenges associated with the inactivation of AMPs, including proteolytic degradation, salt sensitivity, and binding to host proteins. Strategies such as encapsulation, nanoparticle carriers, liposomes, and micelles ensure the successful transport of AMPs to specific sites within microbial cells, such as the inner membrane, where their antimicrobial activity can be exerted. Next, the genus/species of bacteria used in AMP research are important in defining the clinical relevance/importance of the AMP under investigation. Descriptions of research into AMPs (including novel AMPs) and biofilm formation are relatively common and therefore may not have as much scientific impact as the categories mentioned previously. The final two categories included review-type publications and miscellaneous articles that were not easily placed into the categories mentioned above. Finally, although alternative weightings may generate slightly different categorizations, the general description and conclusion of this comprehensive review would not be seriously affected if a few publications switched categories. Most importantly, this categorization process allows the different types of publications to be sorted into more easily describable narrative features, simplifying the narrative review for the reader.

In all, 150 references included research into a total of approximately 150 different AMPs, with these AMPs being obtained from various natural and synthetic sources (Appendix A). Short descriptions and contexts for all of these publications are shown in Section 2 below.

## 2. Results and Discussion

### 2.1. AMP Combination

Zhong et al. investigated novel beta-Ala-modified analogues of anoplin (from solitary spider wasps) [11]. Ano-1β as well as Ano-8β showed synergistic effects against *Pseudomonas aeruginosa* ATCC 27853 in combination with the antibiotics polymyxin B and rifampicin, as well as an additive effect with gentamicin. AMP cecropin A (CecA) was obtained from the greater wax moth (*Galleria mellonella*) and eradicated uropathogenic *Escherichia coli* (UPEC) biofilms, destroying both sessile and planktonic UPEC cells. When combined with the antibiotic nalidixic acid, synergistic activity was observed against UPEC strain CFT073 [12].

Periodontitis and caries are major diseases of the mouth and teeth that require the attention of dentists. Garcia de Carvalho et al. used chlorin-e6 (an extract from chlorophyll obtained from *Spirulina maxima*) conjugated to LL-37 in a nanoemulsion nanocarrier to inhibit biofilms that comprised multiple species during photodynamic therapy [13]. A combination of AMP GH12 with fluoride (GH12/NaF) was found to inhibit the development of caries, regulate the microbiota, and suppress both acid and exopolysaccharide production by biofilms [14]. With respect to the food industry, Gao et al. report using a combination of starch/poly (butylene adipate-co-terephthalate) (PBAT) together with AMPs ε-PL and nisin in an antimicrobial film. This film effectively inhibited *Staphylococcus aureus* and *E. coli* growth and prolonged the shelf life of fresh peaches. This type of approach may be preferable to consumers compared to the use of chemical preservatives in foodstuffs themselves [15]. In hospitals, *Klebsiella pneumonia* is a bacterium associated with difficult to treat multidrug-resistant (MDR) and extensively drug-resistant (XDR) infections. However, AMP K11 generated synergistic effects against this bacterium in combination with ceftazidime, chloramphenicol, meropenem, and rifampicin but not colistin antibiotics. K11 also exhibited good stability in physiological salts and serum, as well as high pH and high thermal stability [16]. Another AMP (Pt5-1c) showed synergistic activity in combination with traditional antibiotics against MDR bacteria including *K. pneumoniae*, *S. aureus*, and *E. coli* [17], while a combination of acidic sophorolipid nanoparticles and LL-37 peptides severely damaged the cell membrane of *E. coli* as shown by atomic force microscopy (AFM) and effectively killed *E. coli*, *S. aureus*, *Staphylococcus epidermidis* and *Pseudomonas aeruginosa* [18]. Finally, combinations of the frog skin AMP temporin L (TL) with different anionic cyclodextrins (types of polysaccharides) have been shown to be potent and safe agents against sessile bacteria [19].

New combinations of AMPs with new or existing antimicrobials or materials could potentially generate killing and anti-biofilm synergies applicable in dental, food, and hospital environments. 

### 2.2. Delivery System

A focal point of treating biofilms lies in optimizing the AMP delivery system used, with 9 out of 25 publications in this section of the manuscript relating to delivery systems centred on carrier particles such as nanoparticles, liposomes, and micelles. The use of nanoencapsulation is exemplified via the encapsulation of the AMP octominin with chitosan (CS) and carboxymethyl chitosan (CMC), with this encapsulation showing greater antimicrobial activity against *Acinetobacter baumannii* and *Candida albicans* compared to octominin alone [20]. Further advancements in this domain include sonochemically synthesized hybrid poly(sulfobetaine) methacrylate/polymyxin B nanoparticles (pSBMA@PM) coated on silicone catheters that synergistically inhibited nonspecific protein adsorption and reduced *P. aeruginosa* biofilm by 97% [21], as well as co-assemblies between host (MSNLP@PEICD) and guest (MagNP@MSNA-CB[6]) mesoporous silica particles that together were able to release the AMP melittin (large molecular weight) and the antibiotic ofloxacin (low molecular weight) when subjected to a pathogen cell stimulus and alternating magnetic field heating. This combination generated ‘strong antibiofilm capacity’ [22]. Moreover, Ali et al. suggested that Poly(lactic-co-glycolic) acid (PLGA) nanoparticles ‘drastically increased’ the SAAP-148 selectivity index (cytotoxicity to antimicrobial activity ratio), by 20-fold against AMR *A. baumannii* and 10-fold against AMR *S. aureus* [23]. The use of graphene-silver nanocomposites (rGOAg) enhanced the delivery of poly-l-lysine-functionalized rGOAg, which efficiently disrupted *S. aureus* biofilm via a ‘contact-kill-release’ mechanism, facilitating bacterial cell membrane disruption and a sudden release of intracellular DNA and proteins [24]. In a separate study, also using silver, Xu et al. reported the synthesis of a AMP/polydopamine/silver nanoparticle nanocomposite (AMP@PDA@AgNPs), which exhibited ‘superior’ activity against both *S. aureus*, *E. coli*, and *P. aeruginosa* compared to AgNPs or AMP alone [25]. Other types of carrier particles such as micelles and liposomes have also shown great potential, as micelles formed from pH-responsive polymer 2,3-dimethylmaleic anhydride-polyethyleneimine-polylactic acid-glycolic acid copolymer (DPP) and photosensitizer chlorin e6 linked to the AMP sequence ‘IRVKIRVKIRVKIRVK’ demonstrated 90% biofilm inhibition in vitro and rapid wound healing in vivo within 15 days when used with laser irradiation (to accelerate wound healing) [26]. Alternatively, liposomes (~280 nm diameter) incorporating the synthetic AMP mimic 7e-SMAMP led to a 75% reduction in *P. aeruginosa* and complete eradication of *E. coli* and *S. aureus* biofilm in an in vitro model. A substantial (~30%) reduction in the inflammatory response of murine macrophages was also observed [27]. Further results in the same area include research using micelles containing a chimeric antimicrobial lipopeptide (DSPE-HnMc) with amphiphilic biodegradable polymers that could to kill a ‘wide spectrum’ of bacteria and their biofilms, including in mouse models [28].

New research into the stability/release of AMPs was available via the co-delivery of nitric oxide (NO) and AMPs crosslinked with hyaluronic acid and divinyl sulfone in an injectable, biocompatible nanogel with sustained 24-h NO release and enhanced antibacterial/antibiofilm activity compared to free NO [29]. Additionally, poly(vinylpyrrolidone) (PVP) microneedle patches were loaded with IR780 iodide and AMP W379s and subsequently coated with 1-tetradecanol. The resultant near infrared (NIR) light-responsive microneedle patches were successfully tested in ex vivo (patient) and in vivo (methicillin-resistant *S. aureus* (MRSA)-diabetic mouse infection model) [30]. Working to improve protegrin-1 (PG-1) stability in physiological fluids, Maystrenko et al. [31] designed the hybrid peptide SynPG-1. SynPG-1 was able to retain activity in 25% serum for up to 24 h, compared to 2 h for the original PG-1 AMP, with MRSA biofilm being prevented by both AMPs. Zhang et al. described a simple strategy for the surface-active factor functionalization of biomaterials using composite biofilms of graphene oxide (GO), poly(lactide-co-glycolide) (PLGA), AMP ponericin G1, and basic fibroblast growth factor (bFGF) [32]. This composite increased the rate of in vivo wound healing and antibacterial effects. The delivery of a custom-designed AMP (W379) via a Janus-type antimicrobial dressing with dissolvable microneedle arrays was found to generate more efficient delivery than a free AMP. Notably, the dressing completely removed dual species MRSA and *P. aeruginosa* biofilm from an ex vivo human skin model [33]. The AMP E6 was used by Lu et al. in combination with the biocide tetrakis hydroxymethyl phosphonium sulfate (THPS) to inhibit the biocorrosion of EH36 ship steel [34]. Although E6 alone was not biocidal and could not prevent the binding of *Desulfovibrio vulgaris*, the AMP appeared to enhance the bactericidal effect of THPS.

Research into AMP-based coatings and implant publications were also quite popular, including a combinatorial approach to antibiotics by developing a method to immobilize nisin on NO-releasing silicone rubber (SR-SNAP-Nisin), which exhibited in vitro anti-infection activity against planktonic and adherent-state *S. aureus* and *E. coli* [35]. Elastin-like recombinamers (ELRs) were used by Acosta et al. to develop an extracellular matrix-mimicking system for anchoring AMPs (e.g., D-GL13K) covalently to implant coatings with strong antibiofilm activity [36]. Mesoporous titania-covered titanium implants were used to deliver RRP9W4N, generating the sustained release of the AMP with equivalent or greater antibiofilm properties than the cloxacillin antibiotic. An additional in vivo study using a rabbit tibia model indicated no negative effects on osseointegration [37]. Immobilisation of the novel AMP JIChis-2 on the Ti-6Al-4V alloy resulted in a functional material with antibiofilm capacities [38], while laterosporulin-coated titanium generated resistance against *S. aureus* biofilms [39]. De Luca et al. impregnated polymer polydimethylsiloxane (PDMS) with r(P)ApoBLPro—an AMP associated with human apolipoprotein B [40]. The stability, peptide release and biocompatibility were evaluated with 70% of the AMP being released after 400 min and biofilm formation by *E. coli* ATCC 25922 being prevented.

Several publications described new approaches to AMP (conjugate) synthesis for AMP delivery. Yang et al. reported the design of a novel neolignan isomagnolone and isomer and their conjugation to form AMP-mimic conjugates (III5 and III15), which exhibited in vitro and in vivo anti-MRSA activity comparable with vancomycin [41]. Cholesterol-modified AMP PMAP-37(F34-R), yielded Chol-37(F34-R), with increased hydrophobicity, thereby potentially increasing bacterial membrane damage and AMP stability. Subsequently, in an *S. aureus* (ATCC 25923) mouse model of peritonitis, Chol-37(F34-R) reduced organ injury and the burden of bacteria [42]. Finally, the potential of ‘click’ chemistry to generate covalent AMP-ionic liquid-based conjugates was described by combining a classical imidazolium IL with the N-terminus of AMP 3.1-PP4 (MeIm-3.1-PP4). This approach maintained the AMPs’ antibacterial activity for a range of pathogenic bacteria, as well as the AMP conjugate’s antibiofilm proliferation activity against an MDR clinical isolate of *K. pneumoniae*, thereby demonstrating improved stability towards tyrosinase-mediated AMP modifications (of relevance, for example, in skin wounds) [43].

Efficient AMP delivery and release is essential in helping target bacteria and biofilms at active concentrations, while helping protect AMPs from rapid degradation. Encapsulation in particles, advances in coatings, and novel methods for AMP (conjugate) synthesis may help to achieve this increased efficiency.

Finally, Milosavljevic et al. designed multifunctional ‘self-propelled microrobots’ (AMP-nanoarchitectonics) that could eradicate MRSA biofilms and aid the delivery and release of drugs [44].

### 2.3. AMP Resistance

Between January 2020 and September 2023, five publications relating to AMP resistance were published, relating to four different biofilm-forming bacteria. Specifically, a mutation of the ‘biofilm-regulating *Francisella* protein Regulator’ (*bfrp*) gene resulted in phenotypic changes indicating that bfrp is a positive regulator of AMP resistance (AMP sheep AMP SMAP-29 and human cathelicidin peptide LL-37) and a negative regulator of biofilm formation in *Francisella novocida* [45]. Lee et al. investigated the induction of the two-component system pcfFeg-pcfRK, the primary mechanism for nisin resistance in *Streptococcus* sp. A12. The results indicated that a novel set of genes (e.g., dipeptidase-PrsW-like protease, Ccma ABC transporter, and Man-PTS) was responsible for nisin resistance [46]. With respect to *Listeria monocytogenes*, a transposon library was used to identify transposants with significantly altered biofilm production levels when compared to a wild-type strain. Inactivation of the phosphatidylglycerol lysyltransferase gene *mprF* led to enhanced biofilm production but increased sensitivity to gallidermin (a type-A-I lantibiotic from *Staphylococcus gallinarum* with some structural similarity to nisin) [47]. Interestingly, Hofer et al. showed that polyunsaturated fatty acids (PUFAs) could generate modest protection or vulnerability to polymyxin B and colistin in *Aeromonas salmonicida*. They concluded that diverse, strain-specific responses to exogenous PUFAs may be an adaptive survival strategy in bacteria that fluctuate between fish and environmental niches [48]. Finally, *Bordetella pertussis* polysaccharide (Bps) was shown to resist AMP (LL-37, polymyxin B, HNP-1, and HNP-2) killing, helping ‘shield’ non-pathogenic K12 *E. coli* and increase in vivo bacterial survival in the respiratory tract [49].

### 2.4. Microbial Focus

AMP-related research into both bacterial and fungal species associated with biofilm production continued in 2020–2023, covering a variety of aspects where the pathogen was the focus of AMP research. In this respect, the pathogens *S. aureus* (25%), *P. aeruginosa* (13%), and *C. albicans*/spp. (13%) were most frequently cited in AMP research (Table 1).


*Enterococcus faecalis*


Three publications specifically focused on *E. faecalis*, with Li et al. investigating the activity of AMP GH12 on this species. GH12 was found to inhibit biofilm formation and virulence, as well as killing these bacteria in an ex vivo tooth model [50]. Staying in the field of dentistry, Mergoni et al. used two antifungal peptides, i.e., KP and L18R, to demonstrate impairment in the biofilm architecture of *E. faecalis* on hydroxyapatite disks [51]. Finally, the cationic AMP lysozyme also appeared to reduce viable *E. faecalis* in biofilms (more extracellular DNA and dead bacteria) [52].


*Escherichia coli*


Enterotoxigenic *E. coli* (ETEC) are responsible for large financial losses for pig farmers. Synthetic AMP GW-Q4 derivative Q4-15-1a exhibited dose-dependent anti-MDR ETEC activity in time–kill kinetic experiments. The minimum biofilm eradication concentration was found to be four-fold the minimal inhibitory concentration (MIC) and in an porcine intestinal epithelial cell model completely inhibited ETEC adhesion [53].


*Pseudomonas aeruginosa*


Four publications focused on human defense peptides such as S100A12, human beta defensins (HBD)-2 and -3, and the cathelicidin-related peptide CRAMP in *Pseudomonas* spp., which inhibit biofilm formation via different mechanisms. Whereas S100A12 inhibited the expression of genes involved in biofilm formation and the synthesis of the virulence factors pyoverdine and pyocyanin [54], the AMPs HBD-2 and -3 had no effects on biofilm-related gene expression but instead induced changes in the *P. aeruginosa* outer membrane that hindered the transport of biofilm precursors into the extracellular space [55]. Stably infected Caco-2 intestinal cells expressing HBD-2 and HBD-3 showed antibiofilm activity against *P. aeruginosa*, indicating that therapeutic strategies that enhance endogenous cellular AMP production might be successful as antibiofilm therapy [56]. The treatment of *P. aeruginosa* with CRAMP led to the dispersion of established biofilms by affecting exopolysaccharides, as well as promoting the flagellar motility of the bacteria inside the biofilm [57]. Yasir et al. generated melimine, a chimeric AMP from melittin and protamine, which inhibited biofilm formation by >75% at one-fold MIC by depolarizing the cell membranes of biofilm cells. Melimine, however, was only effective in eradicating established biofilms when combined with ciprofloxacin (≥61% at one-fold MIC). Importantly, while exposing *P. aeruginosa* to these AMPs or ciprofloxacin at sub-effective concentrations for more than 30 days increased the MIC for ciprofloxacin by 64-fold, the MIC for the AMPs remained unchanged, suggesting that no resistance will be developed to these peptides [58]. Jelleine was first isolated from the royal jelly of bees, with novel analogues being developed by Zhou et al., indicating almost no toxicity yet ‘potent’ inhibition of biofilm formation against MDR *P. aeruginosa* [59]. Established *P. aeruginosa* biofilms in cystic fibrosis possess extracellular DNA that can be degraded by Gaduscidin-1, thereby helping eliminate existing biofilms. This metal-AMP can also be used synergistically with ciprofloxacin and kanamycin [60]. Yin et al. investigated the antimicrobial activity of in silico designed DP7 against 104 clinical *P. aeruginosa* isolates (57 of which were MDR). The AMP was able to inhibit growth, reduce biofilm formation in vitro, and provide a 70% protection rate and 50% reduction in bacterial colonization in chronic biofilm mouse infection models [61].


*Staphylococci*


Most publications focused on *S. aureus*, one of the major causes of skin, gastrointestinal, respiratory tract, and blood stream infections. Additionally, biofilms formed by *S. aureus* are among the most frequent causes of catheter-related or implant-related infections. Human AMP LL-37 appeared to require a ‘quite high’ antimicrobial concentration with MIC values of 132.3 mg/L and 89.6 mg/L for MRSA and methicillin-sensitive *S. aureus* (MSSA), strains, respectively [62]. As previously mentioned for *P. aeruginosa*, Fusco et al. showed that stably infected Caco-2 intestinal cells expressing HBD-2 and HBD-3 also show antibiofilm activity against *S. aureus*, indicating that therapeutic strategies that enhance endogenous cellular AMP production might be successful as antibiofilm therapy [56]. Mutation of the innate defense regulator peptide (IDR-)1018 (1018M) showed activity against MRSA with a 78.9% reduction in biofilm compared to the original peptide and increased binding to the ppGpp stringent response molecule by 27.9% [63]. From non-human sources, the milk-derived AMP BCp12 was shown to bind to accessory gene regulator (agr) QS proteins of *S. aureus*, potentially useful in the food industry [64]. MPX from wasp venom showed activity against *S. aureus* and in a skin scratch model was found to reduce *S. aureus* colonization, thereby resulting in decreased wound size, the promotion of wound healing, and a reduction in inflammation [65]. vCPP2319, a polycationic peptide derived from the Torque virus capsid protein, exhibited activity against established biofilms by killing biofilm embedded bacteria but had no effect on the EPS matrix [66]. The frog skin-derived AMP brevinine-1E-OG9, its analogue (brevinine-1E-)OG9c-De-NH2 [67], and temporin G [68] as well as the synthetic AMP SAMP-A4-C8 not only effectively inhibited and eradicated biofilms but also killed the dormant persister cells, herewith counteracting the recalcitrance of *S. aureus* infections [69]. Both AP7121, produced by *E. faecalis* CECT7121 [70], and in silico designed 1018-k6 [71] dose-dependently inhibited biofilm formation on inert surfaces. Furthermore, the AMPs temporin A, citropin 1.1, and CA(1-7)M(2-9)NH2 were shown to eradicate established biofilms formed on either polystyrene surfaces or central vein catheters [72]. Finally, in two interesting studies, Seo et al. demonstrated that AMPs (HG-1 and haloganan) can be degraded into fragments within hours by the protease aureolysin, secreted by *S. aureus* bacteria in biofilms to shield themselves from treatment [73], while Liu et al. performed in vitro research involving *S. epidermidis*, showing that many isolates could potentially kill competing nasal microbiota by promoting the production of the AMPs HBD-3 and LL-37 while themselves forming resistant biofilms [74].


*Streptococci*


A range of Streptococci were investigated including the major cariogenic species. Dental caries is an oral disease associated with microbial biofilm. The pH in these biofilms is low (pH5.5) due to the metabolic activity of the cariogenic bacteria. Lin et al. conjugated the phototherapeutic agent protoporphyrin IX to the synthetic AMP P12 to enable antimicrobial activity at acidic pH [75]. Honey-derived exosome-like extracellular vesicles (HEc-EVs) were shown to contain a number of AMPs (defensin-1, jelleine 3, and MRJP1) as cargo molecules and bear antibiofilm capacity. Antibiofilm activity was stronger to *S. mutans* than *S. sanguinis* and related to the induction of membrane damage [76]. Zhao et al. incorporated the AMP nisin into a single bond universal adhesive (at 3% *w*/*v*), which could inhibit the growth of *S. mutans* monospecific biofilms, as well as that of saliva-derived multispecies biofilms. Reduced biofilm growth resulted from an inhibitory effect on EPS synthesis and/or excretion [77]. *S. mutans* growth was also inhibited by the strong membrane disrupting activity of the lactotransferrin-derived AMP LF1 [78]. Boswell et al. reported on the effect of lysozyme, lactoferrin, and LL-37 on *S. pneumoniae* serotype 23F, which is associated with respiratory infections. Although a combination of all three AMPs inhibited biofilm-derived bacterial activity, they did not affect the growth of persister cells [79].


*Fungi*


The majority of publications on AMP as treatment for fungi focused on *C. albicans*, which can cause relatively simple to life-threatening invasive infections. The formation of biofilms on tissues and medical devices makes these infections more difficult to treat as they have become less susceptible to antifungal agents. Derivatives (specifically AMP3) of the innate immunity defense protein BPIFA1 inhibited *C. albicans* biofilm formation and downregulated several virulence genes [80]. Housefly larval AMP phormicin C-NS was modified using benzoic and sorbic acid (food preservatives) and inhibited *C. albicans* bud-to-hyphal transition, as well as biofilm formation [81]. Cnt[15-34], the C-terminal part of the AMP Crotalicidin derived from rattlesnake venom, inhibited biofilm formation by disrupting the plasma membrane of *C. albicans* reference strains and flucanozole-resistant isolates [82]. The scorpion venom-derived AMP ToAP2 dose-dependently inhibited the early phase of biofilms (by 60–99.85% at 25–100 μM) [83]. ToAP2 also decreased the viability of mature biofilms but only at 200 μM. Likewise, CGA-N9-C8, an analogue of CGA-N9, also demonstrated strong activity against early phase and established biofilms [84]. Interestingly, this AMP also showed activity against the persister cells of *C. albicans*. Two publications tested AMPs against *Cryptococcus neoformans*, which is frequently detected in immunocompromised individuals and can cause meningitis in HIV infected persons. AMP-17, a novel AMP derived from *Musca domestica*, showed robust activity against both early phase and established biofilms, inhibiting 80% of biofilm formation (BIC80) at concentrations ranging from 16–32 µg/mL (i.e., two–four-fold MIC) and eradicating 80% of the biofilm (BEC80) at concentrations of 64–128 µg/mL, surpassing the efficacy of fluconazole [85]. AMP-17 was also found to inhibit yeast-to-hypha transition and interfered with the adherence of biofilm cells in *C. albicans.* This activity was associated with the mitogen-activated protein kinase (MAPK) pathway and filamentous growth [86]. Snail-derived (*Pomacea poeyana*) AMPs pom-1 and pom-2 reduced the viability of a range of *Candida* spp., including *C. albicans*, *C. auris*, and *C. parapsilosis* [87]. AMP analogues from wasp peptide toxins, MK58911-NH2 and MH58911-NH2, showed strong antimicrobial activity against extra- and intracellular *C. neoformans* [88]. However, only MK58911 possessed activity against early phase and established biofilms, in the 1–8 ug/mL range. The well-known AMPs LL-37 and lysozyme showed no activity to *C. tropicalis* isolates [89].


*Other species*


With respect to systemic human infections, two papers tested AMPs on (carbapenem-resistant) *A. baumannii*, which is a major threat to immunocompromised patients, especially patients in intensive care. The AMP cecropin Cec4 not only inhibits biofilm formation but also eradicates established biofilms of (carbapenem-resistant) *A. baumannii* by disrupting their structure, exhibiting a minimal biofilm inhibitory concentration (MBIC) of 64–128 μg/mL and a minimal biofilm eradication concentration (MBEC) of 256–512 μg/mL [90]. Jung et al. screened 58 AMPs for activity against *A. baumannii* and found two derivatives of the TP4 peptide, i.e., dN4 and dC4, to have therapeutic activity in vivo in mice with *A. baumannii* pneumonia and to eliminate *A. baumannii* biofilms [91]. Hiyash et al. investigated a variety of natural and synthetic AMPs including Mag2, NK2A, and seven synthesized Mag2-derived AMPs (Mag2-17base, 17base-Aib, 17base-Ac6c, 17base-Hybrid, Block, Stripe, and Random). Three synthesized AMPs, i.e., 17base-Ac6c, 17base-Hybrid, and Block, had activity against *Mycoplasma pneumoniae* [92]. The screening of derivatives from the AMP D51 against *Bacillus cereus* (a food poisoning bacterium) highlighted the peptides D51-P11G and D51-P11K for their anti-spore germination and anti-biofilm activities, which they instigated via membrane lysis [93]. Peptide variants of BmKn-2 from scorpion venom exhibited anti-biofilm activity against *Salmonella* isolates [94]. With respect to oral health, Jiang et al. showed that AMP DP7 inhibited planktonic and biofilm forms of *Porphyromonas gingivalis* [95]. MPX, an AMP from wasp venom, was shown to be effective against *Actinobacillus pleuropneumonia*, a highly contagious pathogen causing significant economic loss to the pig industry [96]. The antibacterial activity was not affected by high temperature or low pH, and the AMP protected mice from a lethal dose of *A. pleuropneumonia*. Notably, this same AMP was also active against *S. aureus* [65]. Associated with the food industry, Tanhaeian et al. generated a chimeric peptide (cLFchimera) that included the peptides lactoferricin and lactoferrampin. This AMP was recombinantly introduced into a food-grade strain of *Lactococcus lactis*, with the expressed cLFchimera peptide showing weak (*E. coli*) and strong (*E. faecalis*, *P. aeruginosa*, and *S. aureus*) activity in inhibiting biofilm formation (although the activity was described as ‘moderate’ against *E. faecalis* and *E. coli* in the Discussion) [97]. Finally, B-AMP is a freely available, curated 3D structural and functional repository of biofilm-relevant AMPs. As an example, Mhade et al. demonstrated the use of the repository for developing in silico molecular docking models for the Sortase C Protein of the emerging pathogen *Corynebacterium striatum* [98].

### 2.5. AMP Type

#### 2.5.1. Single AMP Type

The human cathelicidin LL-37 was one of the popular AMPs to be studied during the review period, with LL-37 being one of the few human bactericidal peptides with potent anti-staphylococcal activity (one of the main bacteria associated with orthopeadic implant infections). Using in vitro studies and static biofilm models, Wei et al. showed that LL-37 had significant anti-staphylococcal effects and a ‘destructive effect’ on *S. aureus* biofilm formed on a titanium alloy surface (as a proxy for a prosthesis) [99]. Wuersching et al. examined the effects of LL-37 and human lactoferricin AMPs on anaerobic biofilms associated with oral diseases [100]. This was performed in combination with the antibiotics amoxicillin, clindamycin, and metronidazole using facultative and obligate anaerobic polymicrobial biofilms. The inclusion of LL-37 or lactoferricin enhanced the reduction in both obligate and facultative anaerobic biofilms by these antibiotics. Synthetic AMP TC19 is another human AMP studied, with it being derived from Thrombocidin-1-derived peptide L3 found in human blood platelets. TC19 killed *Enterobacter cloacae*, *Enterococcus faecium*, *S. aureus*, *K. pneumonia*, and *A. baumannii.* Topical application of TC19-containing hypromellose gel significantly reduced MDR *A. baumannii* and MRSA in a murine superficial wound infection model [101]. Another human AMP studied was ‘Reactive Oxygen Species Modulator 1′ (Romo1). Romo1 was observed to eradicate entero-invasive *E. coli* from HeLa cells and a single dose of AMPR-11 (derived from Romo1) appeared to have similar efficacy to multiple doses of imipenem in a mouse model of sepsis when using *A. baumannii*, *P. aeruginosa*, *K. pneumoniae*, and *S. aureus* [102]. As well as humans, insects are also a source of AMPs as Van Moll et al. utilized 36 synthetically produced black soldier fly (*Hermatia illucens*) AMPs. Two cecropins, Hill-Cec1 and Hill-Cec10, were found to be bactericidal with promising activity against *K. pneumoniae* and MDR *P. aeruginosa*. However, although able to prevent *P. aeruginosa* biofilm formation, they could not eradicate existing biofilms [103]. Continuing with a sea life theme, small peptides of 10–22 aa length (Bip_AA_2 and Bip_AA_5) were associated with a significant decrease in the production of biofilm by *Klebsiella oxytoca*. Bip_AA_2 and Bip_AA_5 were obtained via cDNA expression from the Cnidarian moon jellyfish *Aurelia aurita* [104]. Sivakamavalli et al. purified and characterized an AMP crustin from the green tiger shrimp *Peaneaus semisulcatus* [105]. This crustin showed antibacterial activity against Gram-positive *Bacilllus thuringienisis* and *B. pumilis* when compared to Gram-negative *Vibrio parahaemolyticus* and *V. alginolyticus*. From the octopus (specifically *Octopus minor*), the AMP octopromycin inhibited the swarming, swimming, and alginate production of *A. baumannii* via anti-quorum sensing activity. The AMP also significantly reduced the mass and structure of *A. baumannii* biofilm and killed persister cells [106]. Investigation of the AMP capitellacin from the marine polychaeta *Capitella teleta* indicated that this AMP appeared to have a ‘detergent-like’ action on target cells rather than binding to a particular cellular target. Furthermore, after 21 days of exposure, capitellacin did not generate resistance in *E. coli* with the AMP also preventing and destroying mature *E. coli* biofilm [107]. Finally, with respect to aquatic life, Piscidin AMPs are derived from several species of fish with Zhang et al. investigating six analogues of Oreoch-2 that were derived from tilapia (*Oreochromis niloticus*) [108]. This species ranks as one of the most farmed fish species globally. Two of these analogues (ZN-5 and ZN-6) were bactericidal and could inhibit biofilms of *S. aureus* ATCC25923. Prior et al. studied Piscidin AMPs (piscidin 1, piscidin 3, piscidin 4, and piscidin 5) from various bass species of fish. Class II piscidins (piscidin 4 and piscidin 5) were found to be more potently active against *Flavobacterium columnare* and *E. coli* than class I Piscidins (piscidin 1 and piscidin 3), but class I Piscidins showed more growth inhibition of *Aeromonas* spp. [109].

With respect to synthetic derivatives of AMPs, Ajish et al. developed the new symmetric-end AMPs Lf6-pP and Lf6-GG (based on RRWQWRzzRWQWR and centered on D-Pro-Pro) that exhibited potent antibacterial activity against *S. aureus* (KCTC 1621) and *E. coli* (KCTC 1682). Lf6-pP was effective in inhibiting biofilm formation and eradicating mature biofilms of MDR *P. aeruginosa* (MDRPA) strain (CCARM 2095) [110]. The antibiofilm activity of a previously designed AMP WLBU2 against *P. aeruginosa* isolates was investigated by Mazihzadeh et al. [111]. WLBU2 significantly inhibited adhesion to a static abiotic solid surface and biofilm formation in all tested *P. aeruginosa* strains. Additionally, exposure to WLBU2 resulted in a >4-fold reduction in biofilm-associated gene (e.g., rhlI, rhlR, lasI, and lasR) expression, while biofilm formation was significantly inhibited in a murine catheter-associated carbapenem-resistant *P. aeruginosa* (CRPA) infection model. ‘Unnatural’ star-shaped AMPs were designed by Pan et al. via homologues of poly(l-lysine)s and poly(l-alpha,zeta-diaminoheptylic acid)s. The star-shaped poly(l-ornithine) PO3 had high proteolytic stability and broad-spectrum antimicrobial activity especially against Gram-negative *P. aeruginosa*. In vivo studies using a *P. aeruginosa*-infected murine skin burn model revealed that PO3 alleviated inflammation and exhibited a wound healing process [112]. Using a different approach, Ghimere et al. reported on ‘the continued synthetic molecular evolution of a lineage of host-compatible AMPs’ using variants derived from an evolved AMP, which was originally derived from a consensus sequence where two invariant glycines had been removed (termed ‘D-amino acid CONsensus with Glycine Absent’ or ‘D-CONGA’). One variant (D-CONGA-Q7), with a polar glutamine inserted into the middle of the sequence, showed a ‘gain of function’ by being active against innately resistant clinical *K. pneumoniae* isolates [113]. Bactenecin from bovine neutrophils is one of the shortest natural broad-spectrum AMPs currently known. By attaching a hydrophobic fatty acid ligand to the AMP Bac8c (RIWVIWRR-NH2), the new LA-Bac8c demonstrated improved antibiofilm activity towards *S. aureus* and MRSA compared to Bac8c [114]. Several publications focused on modifications of the human endogenous AMP LL-37. Laksmaiah Narayana et al. generated a library of lipopeptides via sequential truncation of the AMP KR12 (derived from LL-37), which were also conjugated to fatty acids. They found that the miniature LL-37-like peptide C10-KR8d (d-form) was able to reduce MRSA bacterial burden in a neutropenic mouse model. In addition, C10-KR8d prevented bacterial biofilm formation in an *S. aureus* mouse catheter model [115]. AMP GA-KR12 (a short yet active peptide of LL-37) has novel dual-action activity that inhibits the growth of *S. mutans* biofilm and promotes the remineralization of human dentine blocks with artificial carious lesions [116]. Dimerization and backbone cyclization of KR-12 facilitated the production of the synthetic peptide CD4-PP, which was active against type and clinical uropathogenic strains of *K. pneumoniae*, *P. aeruginosa*, and *E. coli.* CD4-PP reduced *E. coli* attachment to pieces of a urinary catheter placed in saline/CD4-PP fluid [117]. Interestingly, Hacioglu et al. compared the antibiofilm activities of a range of ceragenins (synthetic amphipathic molecules that are designed to mimic naturally occurring AMPs) against AMPs LL-37, magainin, and cecropin. At the concentrations tested, ceragenins (CSA-13, CSA-90) were reported to be much more effective against mono- and multi-species biofilms than AMPs, with them being particularly effective against *C. albicans* biofilms [118]. The efficacy of AMPs as antimicrobial drugs is reduced by their low proteolytic stability, suboptimal bioavailability, and relatively high manufacturing costs. In this respect, Yu et al. described the synthesis of small molecules based on biphenylglyoxamide (15c), which possessed Gram-positive and -negative antibacterial activity and disrupted (35%) of *S. aureus* biofilm [119]. The compounds appeared to be non-toxic. GH12 is an example of an AMP that has been de novo designed and synthesized with an emphasis on anti-dental activity. Jiang et al. investigated the AMP’s effect in a complex biofilm model in vitro and a rat caries model in vivo. GH12 reduced the exopolysaccharide and lactic acid production in a S. mutans biofilm, as well as the abundance of caries-associated bacteria in a rat model and maintained microbial diversity in multispecies dental plaque-derived biofilms obtained from healthy volunteers [120]. In a similar publication, the same lead author investigated the pH responsiveness of GH12, showing that this AMP generated much lower minimal bactericidal concentrations and minimal inhibitory concentrations at pH 5.5 as compared to pH 7.2. The relevance being that dental caries are associated with acidification of biofilms on the affected teeth [121]. Consecutive exposure to sub-lethal levels of GH12 tended not to generate GH12 resistance, although the acquisition of resistance could not be ruled out. TAT-RasGAP317-326 is a cell-permeable chimeric peptide derived from the p120 Ras GTPase-activating protein that has anticancer properties via membrane lysis and cell death. Heinonen et al. showed that this new AMP could limit biofilm expansion and prevent biofilm formation by *P. aeruginosa* and *A. baumannii*, while also inhibiting *S. aureus* biofilms [122]. Lima et al. used ‘Collection of AMPs (CAMPR3)’ software to cleave AMP ILTI obtained from plant seeds, which exhibits biological activity against pest insects. The result was the 198 amino-acid peptide KWI18 [123]. KWI18 reduced bacterial (*P. aeruginosa*) and fungal (*C. parapsilosis*) biofilms.

#### 2.5.2. ‘Novel’ AMP Types

The following publications include the word ‘novel’ in their titles but do not include publications containing ‘novel’ when related to methods or specific bacterial species. Eight publications belong to this category. Of particular note during 2020–2023 is the discovery and testing of novel AMPs from various types of aquatic life. One such example is the marine mud crab *Scylla paramamosain*, with Chen et al. investigating a 22 amino-acid variant of the C-type lectin homolog SpCTL6 (Sp-LECin) [124]. This AMP inhibited the growth of a range of Gram-negative bacteria, e.g., *P. aeruginosa*, *A. baumanii*, and *Shigella fiexneri* and Gram-positive bacteria, e.g., *L. monocytogenes*, *E. faecium*, and *E. faecalis*, as well as showing anti-biofilm maturation and the formation for *P. aeruginosa*. The AMP octoPartenopin was extracted from the suckers of *Octopus vulgaris*. This AMP possessed high protease activity, with an analogue of this AMP showing inhibition and eradication activity against *P. aeruginosa*, *S. aureus*, and *C. albicans* biofilms [125]. A recombinant version of scyreprocin (found in the male gonads of this crab and named rScyreprocin) was identified as an interacting partner of the AMP SCY2. Interestingly, this compound not only possessed antibacterial and antibiofilm activity but also antifungal (*Cryptococcus* spp. and *Candida* spp.) and *Aspergillus* spp. anti-spore germination activity [126]. AMPs have also been isolated from bacteria, e.g., Zhang et al. aimed to simplify the structure of the AMP PE2 isolated from the endospore-forming, facultative anaerobic bacterium *Paenibacillus ehimensis*. Linear analogues of AMP PE2, numbered 26 and 27, showed significant activity against both MDR bacteria in vitro as well as in a mouse pneumonia model [127]. A different AMP, ‘YS12′, was derived from *Bacillus velezensis* strain CBSYS12 via the Korean food kimchi. This showed strong antimicrobial activity against MRSA 4–5, VRE 82, *E. coli*, *P. aeruginosa*, and *Mycobacterium smegmatis* and inhibited 80% of biofilm formation by *E. coli* and *P. aeruginosa* [128]. Using rational design, Masadeh et al. combined the alpha-helical parts of BMAP-28 (bovine cathelicidin) and LL-37 (human cathelicidin), generating the hybrid peptide MAA-41. This hybrid peptide showed strong activity against biofilm-forming cells, as well as synergistic or additive effects in combination with conventional antibiotics [129]. Shang et al. designed AMPs containing ’RWWWR‘ as a central motif together with arginine (R) end-tagging. One AMP (Pep 6) generated a 62–90% reduction in *S. aureus* skin burn and *E. coli* bacteremia models. Moreover, Pep 6 could reduce robust mature MRSA biofilm by 61% [130]. Ramakrishnan et al. designed a novel AMP SS-BF-3, against *P. aeruginosa*, which was developed using machine learning and a random forest algorithm with decision tree. However, the publication did not include in vitro or in vivo data [131]. Finally, Jiang et al. used bioinformatics to generate four truncated AMPs of Spampcin (Spa31, Spa22, Spa20, and Spa14) [132]. Spa31 demonstrated potent antimicrobial activity against *E. coli*, *P. aeruginosa*, and *S. aureus*, inhibited/eradicated *P. aeruginosa* biofilms, and significantly improved the survival rate (from 50% to 79%) of *Danio rerio* zebrafish infected with *P. aeruginosa* [132].

#### 2.5.3. AMP Models

Several AMP publications were associated with interesting in vitro or in vivo models. Chitosan-derived gels loaded with the novel AMPs ASP-1 and ASP-2 were found to be effective on monospecies biofilms grown for three days on ex vivo porcine skin and in vitro (polymer mesh) models. Furthermore, over seven days, 70 to 80% of AMPs were released from the chitosan [133]. Synergistic interactions between species (e.g., *C. albicans* and *S. aureus*) can increase biofilm formation and AMR. Although AMP L18R was able to reduce the biomass of monomicrobial *C. albicans* and *S. aureus* early-stage and mature biofilms, no reduction in biomass was observed against a polymicrobial biofilm in the Lubbock chronic wound biofilm model. However, L18R caused a moderate reduction in total CFU number and decreased the number of *C. albicans* in the same model [134]. A novel murine osteomyelitis model using the infected femoral bone canals of laboratory rats was used by Melicherčík et al. to study bone implant-related infections [135]. A 12 amino acid synthetic analogue of the AMP halictine-2 was mixed into polymethylmethacrylate bone cement, preventing the subsequent formation of MRSA biofilm on the surfaces of more than 80% of these implants. In another interesting murine model, Martinez et al. studied two synthetic AMPs (P5 and P6.2) that had been earlier designed via combined computer-assisted and rational approaches [136]. Both peptides were investigated using a *P. aeruginosa* lung infection model of neutropenic mice, with the lungs being extracted to assess both bacterial load and pro-inflammatory cytokine activity. A significant reduction in bacterial load and a concomitant decrease in pro-inflammatory cytokines IL-1β, IL-6, and TNF-α was observed. The bacterium *Desulfovibrio vulgaris* and a corrosion biofilm model (using steel rivets as a surface for bacterial growth in a 96-well format) was developed by Stillger et al., who showed that AMPs S6L3-33 (bacterial reduction), bactenecin (total biomass reduction), and DASamP1 (biofilm inhibition up to 21 days) were the most useful anti-biocorrosive AMPs of those tested [137]. Finally, AMPs may be important in protecting foodstuffs from spoilage bacteria. In this respect, Zhang et al. utilized three food matrices (cooked meat sauce, raw fish, and fruit juice) to investigate the AMPs As-CATH4, and As-CATH5 and Hc-CATH (from the sea snake *Hydrophis cyanocinctus* and Chinese alligator *Alligator sinensis*, respectively) that had previously been identified to exhibit broad-spectrum antimicrobial activities. Sodium benzoate and potassium sorbate were used as positive controls. Synergistic interactions were observed in combination with butyl paraben in fruit juice [138].

Published research into novel AMPs and their variants included AMPs obtained from a wide variety of (micro)organisms, including humans, mice, aquatic life, insects, plants, and even bacteria. Several interesting models for evaluating AMP efficacy were also published including mesh, bacterial/fungal synergistic, murine, and food preservative models.

### 2.6. Miscellaneous

Seven publications were assigned to the ‘Miscellaneous’ category, including publications relating to the optimization of AMPs (L5K5W, S6L3-33, bactenecin, and DASamP1) against biocorrosive bacteria [139], marine-based AMPs [140], and a real time thermal sensor for quantifying the inhibitory effects of AMPs (protamine and OH-CATH-30) on biofilms and bacterial adhesion [141]. A comment relating to the AMP cecropin A, derived from moths, was made by Fenner [142].

Ravichandran et al. [143] describe a structural and functional repository containing >5000 structural models (including a curated collection of 2502 biofilm protein targets against a total of 473 bacterial species), while Xu et al. [144] provide a comprehensive assessment of machine learning-based methods for accurately identifying AMPs. The original text by Li et al. was written in Chinese, and only the abstract was available in English [145]. From their abstract, AMP GH12 decreased biofilms consisting of *S. mutans* and the commensal bacteria *S. sanguinis* and *S. gordonii*.

Various novel tools, databases, and environments are available to aid researchers interested in AMP and biofilm research.

### 2.7. Review-Type

A total of 15 review-type articles remained in the total list of 150 publications generated by our search strategy, even though the strategy excluded ‘reviews/expo’, ‘conference abstracts’, and ‘conference reviews’. Seven publications covered the topic of (multi-) drug resistance [146,147,148,149,150,151,152], two related to surfaces/devices [153,154], and two focused on ‘therapeutics’ [155,156]. The other two review-types focused on biofilm matrix dynamics [157] and the severity of biofilms in disease [158]. One review-type article focused on the stomach pathogen *Helicobacter pylori* [159], whilst one focused on bacteriocins [160].

Many of these articles could have been included in the categories used to divide the research publications into different themes but are broad in scope and did not provide the type of information required for the current comprehensive review.

## 3. Conclusions

During the period of January 2020 to September 2023, research into the effect of AMPs on biofilms resulted in 150 publications (when using the search criteria indicated in this manuscript). In general, publications involving ‘single AMP types’ and ‘delivery systems’ were most frequent, with research being performed on a wide range of existing and new AMPs, of which LL-37, GH12, and nisin were most frequently cited (Appendix A). This is notable as research into LL-37 and nisin has been published over several decades but still continues to the present day. That said, research into new and modified AMPs continues to be published, although it remains to be seen if these new/modified AMPs will eventually find a substantial place in the prevention/removal of biofilms in clinical, farming, or food applications. With respect to target bacteria, research involving *E. coli* and *E. faecalis* was least popular, with *S. aureus* and *P. aeruginosa* being the most popular microorganisms associated with biofilm and AMP research (Table 1), which is a reflection of the continued importance of these species in medical and veterinary AMR infections. AMP ‘resistance’ continues to be published but at a relatively low level compared to the other categories utilized in this manuscript. This most likely reflects the lack of impact of AMP resistance in clinical, farming, or food applications, which itself is probably associated with the lack of (large-scale) AMP use in these environments.

## 4. Future Outlook

Between January 2020 and September 2023, this review identified research into approximately 150 different AMPs. However, despite decades-long research into AMPs, these compounds are still not widely used in clinical, farming, or food environments. These implementation issues need to be overcome if the use of AMPs is to be successfully applied in the future and include general problems such as AMP susceptibility to protease activity and bacterial AMP resistance mechanisms. With respect to proteolysis, continued research into the development of novel AMP delivery systems that shield AMPs from proteolytic digestion (at least until they reach the intended site of activity) may yield encouraging results. With respect to bacterial AMP resistance, research into new AMPs and new AMP combinations (i.e., AMP + AMP or AMP + other antibiotic compounds) may be a step forward in helping prevent the development and/or spread of (multi-drug) resistance. Research into existing and novel AMPs continues to be published, although for the future, a focus on overcoming the hurdles to AMP implementation in medical, farming, and food environments should be encouraged.

## Figures and Tables

**Figure 1 antibiotics-13-00343-f001:**
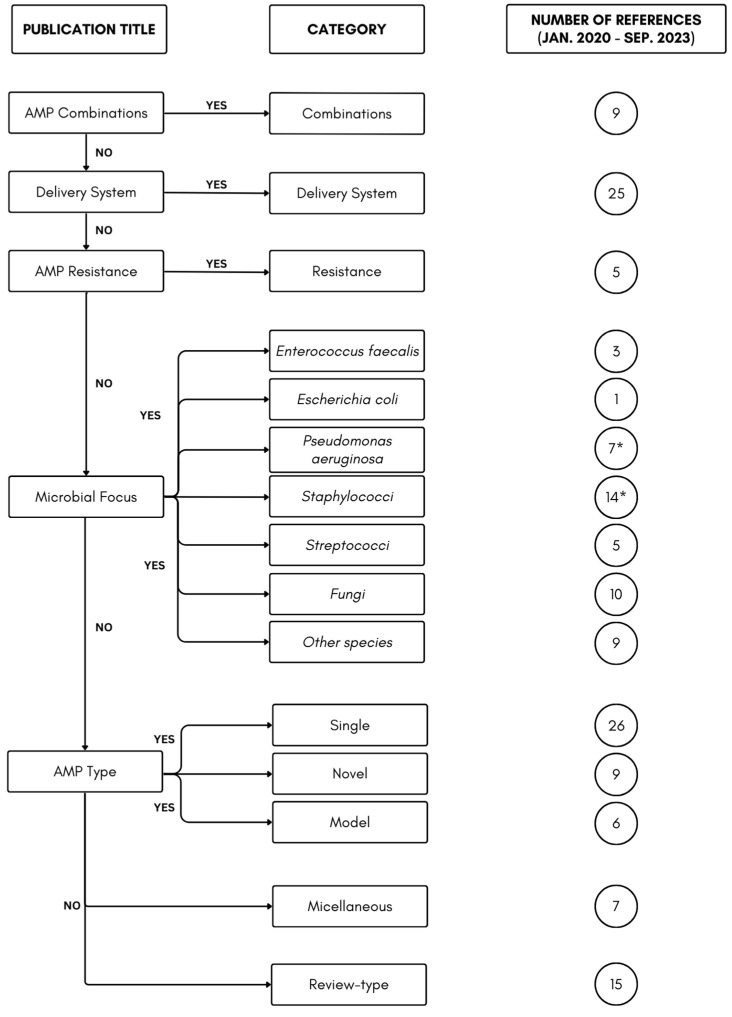
Hierarchy for organizing 150 ‘AMP–and–biofilm’-related publications (January 2020–September 2023) into seven categories. Top = highest weighting. * = One publication specifically mentioned *Staphylococcus aureus* and *Pseudomonas aeruginosa* in its title.

**Table 1 antibiotics-13-00343-t001:** Frequency of microorganisms associated with published ‘AMP and biofilm’ research where the pathogen was the focus of interest in the title (from January 2020 to September 2023). Two publications reported research on *S. aureus*/*S. epidermidis* and *S. aureus*/*P. aeruginosa*.

Organism	Number
*Staphylococcus aureus*/MRSA	13
*Pseudomonas aeruginosa*	8
*Candida albicans*/spp.	8
*Streptococcus pneumonaie*/*mutans*/*oral*	5
*Enterococcus faecalis*	3
*Escherichia coli*	1
*Acinetobacter baumannii*	2
*Cryptococcus neoformans*	2
*Staphylococcus epidermidis*	2
*Actinobacillus pleuropneumoniae*	1
*Bacillus cereus*	1
*Corynebacterium striatum*	1
*Lactococcus lactis*	1
*Mycoplasma pneumoniae*	1
*Porphyromonas gingivalis*	1
*Salmonella serovars*	1

## Data Availability

Not applicable.

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
