# Peer review of "A Comprehensive Review of Recent Research into the Effects of Antimicrobial Peptides on Biofilms—January 2020 to September 2023"

_antibiotics, 2024, doi:10.3390/antibiotics13040343_

Round 1

Reviewer 1 Report

Comments and Suggestions for Authors

This review article encompasses a wide gamut of reports of antimicrobial peptides (AMPs) over the last 4 years. The review has highlighted the wide effectiveness of AMPs against biofilms and the potential of AMPs in a field of antibiotics that has met with increasing multi-drug resistant microorganisms. The review has been organized into a neat hierarchy with focus of delivery systems, AMP resistance and the various ESKAPE pathogens. 

The scope of AMPs covered in this review is extensive showcasing the huge amount of work that has been done in this arena in the last 4 years. However, review articles related to antimicrobial peptides in the last 4 years has been limited and mostly dealt with structural aspects and classifications of these AMPs. That is why I believe that this review fills that gap and should be published.

Author Response

The authors would like to thank the reviewer for his/her comments and acceptance of the manuscript. 

Reviewer 2 Report

Comments and Suggestions for Authors

1. General comment

In the manuscript, the author reviewed a comprehensive review of 150 publications (from January 2020 to September 2023) concerning polypeptide antibiotic agent, antimicrobial peptide and biofilm, presenting promising evidence for the role of AMPs as effective antimicrobial agents.

2. Major revision

1) Supplementary Table S1

As this review focused on AMP research, it is strongly recommended to add the amino acid sequence, molecular mass and isoelectric point (pI) of AMP, especially amino acid sequence, so that the reader of antibiotics can easily and precisely understand the content of the manuscript.

2) Reference

It is essential to check the reference, including pages, volume, year, journal name and so on.

3. Minor revision

1) Lines 412 and 430: It is recommended to explain a word, BEC80, in relation to MBEC.

2) Line 527: Check a word, CONsensus.

Author Response

The authors thank the reviewer for his/her comments.

1) Supplementary Table S1

As this review focused on AMP research, it is strongly recommended to add the amino acid sequence, molecular mass and isoelectric point (pI) of AMP, especially amino acid sequence, so that the reader of antibiotics can easily and precisely understand the content of the manuscript.

 Supplementary Table S1 has been comprehensively updated. "Finding these sequences was in itself a challenge and the authors feel that the many extra hours required to add molecular masses and pI would not add any substantial increased value to the review in its current form for the likely readership. However, we thank the reviewer for mentioning this request

 2) Reference

It is essential to check the reference, including pages, volume, year, journal name and so on.

All references have been comprehensively reviewed and changed where necessary.

 Minor revision

1) Lines 412 and 430: It is recommended to explain a word, BEC80, in relation to MBEC.

The following revisions have been made …

“AMP-17, a novel AMP derived from Musca domestica, showed robust activity against both early phase and established biofilms, inhibiting 80% of biofilm formation (BIC80) at concentrations ranging from 16-32 µg/mL (i.e., 2-4-fold MIC) and eradicating 80% of the biofilm (BEC80) at concentrations of 64-128 µg/mL, surpassing the efficacy of fluconazole.” (lines 412-416).

 and

 “The AMP cecropin Cec4 not only inhibits biofilm formation, but also eradicates established biofilms of (carbapenem-resistant) A. baumannii by disrupting their structure, exhibiting a minimal biofilm inibitory concentration (MBIC) of 64–128 μg/mL and a minimal biofilm eradication concentration (MBEC) of 256-512 μg/mL (90).” (lines 431-435).

2) Line 527: Check a word, CONsensus.

 The following revision has been made …

“Using a different approach, Ghimere et al., reported on ‘the continued synthetic molecular evolution of a lineage of host-compatible AMPs’ using variants derived from an evolved AMP, which was originally derived from a consensus sequence where two invariant glycines had been removed (termed 'D-amino acid CONsensus with Glycine Absent' or ‘D-CONGA’).” (lines 529-533).

Reviewer 3 Report

Comments and Suggestions for Authors Fontanot et al have worked on compiling the currently available research (Jan 2020 to Sept 2023) on the effect of AMPs on biofilm formation. The authors have done a comprehensive search into the literature using the appropriate keywords and funneling strategies to ensure that they are looking at the most relevant information. Additionally the authors do a nice job of addressing the limitations associated with the use of AMPs in the clinic or food industries. Overall the publication is well written.   Minor concerns -    Page 2 Line 91 - Here do the authors mean delivery systems that deliver AMPs to the inner membrane directly? Please add a couple sentences to clarify.

Author Response

The authors thank the reviewer for his/her comments.

  1. Minor concerns -    Page 2 Line 91 - Here do the authors mean delivery systems that deliver AMPs to the inner membrane directly? Please add a couple sentences to clarify.

An extra sentence has been added “Investigations into AMP delivery efficiency are crucial for potentially overcoming challenges associated with inactivation of AMPs, including proteolytic degradation, salt sensitivity, and binding to host proteins. Strategies such as encapsulation, nanoparticle carriers, liposomes, and micelles ensure the successful transport of AMPs to specific sites within microbial cells, such as the inner membrane, where their antimicrobial activity can be exerted.” (Lines 90 – 95).